# Does Decoherence Select the Pointer Basis of a Quantum Meter?

**DOI:** 10.3390/e24010106

**Published:** 2022-01-10

**Authors:** Abraham G. Kofman, Gershon Kurizki

**Affiliations:** Department of Chemical and Biological Physics, Weizmann Institute of Science, Rehovot 761001, Israel

**Keywords:** quantum measurements, decoherence, pointer states, Quantum Lamarckism, the observer in quantum mechanics

## Abstract

The consensus regarding quantum measurements rests on two statements: (i) von Neumann’s standard quantum measurement theory leaves undetermined the basis in which observables are measured, and (ii) the environmental decoherence of the measuring device (the “meter”) unambiguously determines the measuring (“pointer”) basis. The latter statement means that the environment *monitors* (measures) *selected* observables of the meter and (indirectly) of the system. Equivalently, a measured quantum state must end up in one of the “pointer states” that persist in the presence of the environment. We find that, unless we restrict ourselves to projective measurements, decoherence does not necessarily determine the pointer basis of the meter. Namely, generalized measurements commonly allow the observer to choose from a multitude of alternative pointer bases that provide the same information on the observables, regardless of decoherence. By contrast, the measured observable does not depend on the pointer basis, whether in the presence or in the absence of decoherence. These results grant further support to our notion of Quantum Lamarckism, whereby the observer’s choices play an indispensable role in quantum mechanics.

## 1. Introduction

Attempts to banish the observer from quantum mechanics have motivated approaches [1,2,3,4] whereby the *environment observes* a quantum system. These approaches “objectivize” quantum measurement theory by substituting the environment-induced decoherence of a quantum observable for its unread (nonselective) measurement. In the simplest version of these approaches introduced by von Neumann [5], the environment and the system are entangled by their interaction, and the environment is then ignored (traced out), decohering the reduced state of the system. Subsequent theory, notably Zurek’s, has pleaded the case for “the environment as the observer” by stressing the importance of system-environment correlations in determining the information obtainable on the system through the notions of “einselection” [4,6] and “the environment as a witness” [7,8] and the mechanism of enforcing classicality [9,10,11].

These approaches must cope with the issue that *the decomposition of a (closed) “supersystem” into an open quantum system and its environment is often neither unique nor inevitable, but rather a matter of expediency and choice for the observer:* Depending on the computational and experimental resources, the observer can choose which degrees of freedom pertain to the system to be measured (or otherwise manipulated) and which ones are part of the inaccessible environment (“bath”). However, even after this choice has been made, the observer must choose what observable of the system to measure and how frequently. We have long stressed that einselection, which singles out the states of a quantum system that are resilient to decoherence, is restricted to long time scales compatible with the Markovian (memoryless) assumption concerning the environment (bath) response [12]. Conversely, it excludes much shorter non-Markovian time scales that are restricted to the memory or correlation time of the bath response [13], a time scale that is often overlooked.

However, as we have shown, *this division of time scales is invalid* when the quantum system is subject to monitoring by the observer, even if such monitoring is considered non-intrusive, corresponding to quantum nondemolition (QND) measurements that leave the quantum observable of the system intact [12,13]. Nevertheless, the ensuing system–bath dynamics may drastically deviate from the course prescribed by decoherence or dissipation or even from the course prescribed by dynamical control [14,15,16,17,18,19,20].

Such measurement-induced dynamics may steer the system to a final state where it is heated up (in the Zeno regime) or cooled down (in the anti-Zeno regime [21]) irrespective of the bath temperature [22,23]. The observer’s ability to steer the evolution of open systems is the basis for our fundamental approach we have dubbed Quantum Lamarckism [24] whereby the system evolution is dictated not merely by decoherence or bath effects but by its functional adaptation to the observer’s choices.

Here we seek further support for the view embodied by Quantum Lamarckism that the observer cannot be banished from quantum mechanics. We do so by exploring the choices available to the observer in selecting the pointer basis of a meter in the presence of decoherence.

In the standard (von Neumann) quantum-measurement theory [5] an observable of a system S is measured by coupling the system S to a “meter” M and then measuring the latter. Namely, S is observed via M. von Neumann’s theory is moot concerning the choice of basis for M and the effects of decoherence on M. By venturing beyond von Neumann’s theory, Zurek investigated [3,4,6] what happens when the observable (pointer) of M differs from the “standard pointer”, which commutes with the state of M after the S-M interaction, and what are the consequences of decoherence of M. His investigations can be briefly summarized by the following points:(a)The measured observable of S is uniquely determined by the measured observable of M.(b)Decoherence “dynamically selects” the pointer basis of M.(c)As a consequence of (a) and (b), the decoherence of M “dynamically selects” the measurable observables of S and M [3], which leads to Zurek’s notion of einselection [4,6].

Our analysis shows that the pointer-basis selection for a quantum meter in the presence of decoherence is not necessarily restricted by einselection. We find that, unless we restrict ourselves to projective measurements of the observable by the meter, decoherence does not in general select the pointer basis of M (Section 2). Under mild conditions, there is a multitude of alternative pointer bases the observer can choose from, all of which are capable of providing the same information on the observable by means of generalized measurements, regardless of decoherence. By contrast, the selection of the pointer basis of M does not affect the measured observable, which remains unique, whether in the absence or in the presence of decoherence (Section 3). We illustrate these results for the case of a qubit meter decohered by a bath when this meter measures a two-level system (Section 4). These results are discussed as arguments in favor of the central role of the observer in quantum mechanics in the spirit of Quantum Lamarckism [24] (Section 5).

## 2. Quantum Pointer Resilient to Decoherence

Let us consider a measurement of an observable $\hat{S}$ of system S by a meter M that is subject to decoherence by a bath B. Although our analysis can be completely general, we choose for simplicity the S-M interaction (via Hamiltonian $H_{\mathrm{SM}}$) to be much stronger (hence faster) than that of M-B (via Hamiltonian $H_{\mathrm{MB}}$). The measurement process then consists of three distinct stages:

(1) S and M interact over time interval $\tau_{M}$ that is long enough to entangle the two, but short enough to ignore the effects of B. The observable of the system S to be measured is represented in the basis of its (orthonormal) eigenstates $|S_{n}\rangle$, as
(1)$$\hat{S}=\sum_{n}\epsilon_{n}|S_{n}\rangle \langle S_{n}|,$$
whereas the *unknown* initial state of the system S is
(2)$$|\psi_{S}\left(0\right)\rangle =\sum_{n}c_{n}|S_{n}\rangle .$$
The initial factorized state of S and M then evolves over the time interval $\left(0,\tau_{M}\right)$ to a state that obeys the Schmidt decomposition,
(3)$$|\psi_{\mathrm{SM}}\left(0\right)\rangle =\sum_{n}c_{n}|S_{n}\rangle ⊗|M\rangle \;\to \;|\psi_{\mathrm{SM}}\left(\tau_{M}\right)\rangle =\sum_{n}c_{n}|S_{n}\rangle ⊗|P_{n}\rangle ,$$
where the meter states also satisfy orthonormality.

For a proper measurement of the observable $\hat{S}$ we impose the back-action evasion (quantum non-demolition) condition [25],
(4)$$[\hat{S},H_{\mathrm{SM}}\left(t\right)]=0,$$
on the system-meter (SM) coupling Hamiltonian $H_{\mathrm{SM}}\left(t\right)$. Moreover, we assume that $\tau_{M}$ is sufficiently short, so that the system Hamiltonian $H_{S}$ can be neglected during the S-M interaction (the impulsive limit). We also neglect the meter Hamiltonian.

A measurement in the basis of the meter states $|P_{n}\rangle$ collapses the system state to an eigenstate $|S_{n}\rangle$ and thereby yields the eigenvalue $\epsilon_{n}$ of $\hat{S}$ with the probability $|c_{n}{|}^{2}$. The meter observable has then the form
(5)$$\hat{P}=\sum_{n}b_{n}|P_{n}\rangle \langle P_{n}|,$$
which we dub the *standard pointer*, since it effects ideal (projective) measurements of the system. The corresponding meter state, obtained from Equation (Equation 3) upon tracing over S, is then
(6)$$\rho_{M}\left(\tau_{M}\right)=\sum_{n}|c_{n}{|}^{2}|P_{n}\rangle \langle P_{n}|.$$

(2) On a much longer time scale, $t\gg 1/\gamma \gg t_{c}$ where 1/γ is the decoherence time of the meter (M) and $t_{c}$ is the correlation (memory) time of the decohering bath (B) [12,13], we choose a nondegenerate meter variable $\hat{Q}$ which satisfies the back-action evasion condition for the M-B interaction,
(7)$$[\hat{Q},H_{\mathrm{MB}}\left(t\right)]=0,$$
where $H_{\mathrm{MB}}\left(t\right)$ is the M-B coupling Hamiltonian. This condition ensures that the eigenstates $|Q_{n}\rangle$ of $\hat{Q}$ are invariant under decoherence. In order to conform to Zurek’s analysis [3,6,7,8], we take $H_{\mathrm{MB}}$ to commute with $H_{\mathrm{SM}}$. Then the standard pointer $\hat{P}$ of stage 1 can be shown to be identical with $\hat{Q}$. Upon tracing out B, we then arrive at the S-M state that is stationary and diagonal in the bases $\left\{|S_{n}\rangle \right\}$ and $\left\{|Q_{n}\rangle \right\}$ as t→∞,
(8)$$\rho_{\mathrm{SM}}\left(t\right)\;\to \;\rho_{\mathrm{SM}}^{\infty }=\sum_{n}|c_{n}{|}^{2}|S_{n}\rangle |Q_{n}\rangle \langle S_{n}|\langle Q_{n}|.$$
Namely, decoherence eliminates the off-diagonal elements of the joint S-M state and acts as a nonselective measurement without a readout of the measurement results of the meter by the bath. Since now the standard pointer coincides with $\hat{Q}$, this means that the bath performs a non-selective measurement of the system.

(3) At stage 3, which follows the decoherence stage 2, projective measurement of the meter is performed on $\rho_{\mathrm{SM}}^{\infty }$ in the $\left\{|Q_{n}\rangle \right\}$ basis. This measurement is assumed to be fast (impulsive), so that the evolution of the meter during the measurement may be neglected. Then, a measurement of $\hat{Q}$ yields a selective projective measurement of the observable $\hat{S}$. Namely, an eigenvalue $\epsilon_{n}$ of $|S_{n}\rangle$ is obtained with probability $|c_{n}{|}^{2}$.

These results adhere to Zurek’s view regarding the pointer basis [3,6,7,8]: They show that decoherence determines a meter state that is diagonal in the basis $\left\{|Q_{n}\rangle \right\}$, and only this basis can yield projective measurements of the system. Decoherence dynamically selects a unique “resilient” basis $\left\{|Q_{n}\rangle \right\}$, whereas any pointer basis differing from $\left\{|Q_{n}\rangle \right\}$ cannot yield projective measurements of the system. The question we raise is: Does this advantageous property single out $\hat{Q}$ as the only appropriate pointer?

## 3. Alternative Quantum Pointers

To answer this question, consider the general case where the standard pointer $\hat{P}$ does not commute with $\hat{Q}$, which is invariant under the action of $H_{\mathrm{MB}}$. This means that $\hat{P}$ and $\hat{Q}$ are determined independently, by the non-commuting Hamiltonians $H_{\mathrm{SM}}$ and $H_{\mathrm{MB}}$. Moreover, we consider a selective measurement of a meter variable $\hat{R}$ arbitrarily chosen by the observer. Generally, $\hat{R}$ commutes neither with $\hat{P}$ nor with $\hat{Q}$. Whereas in von Neumann’s theory the S-M correlation (stage 1) is directly followed by a selective measurement of the meter (stage 3), we here adopt Zurek’s procedure whereby stage 3 is preceded by a nonselective measurement of the meter caused by decoherence (stage 2)

At t→∞, i.e., after the completion of decoherence, we then have
(9)$$\rho_{\mathrm{SM}}\left(\infty \right)=\sum_{k,l,n}c_{n}c_{l}^{*}\langle Q_{k}|P_{n}\rangle \langle P_{l}|Q_{k}\rangle \;|S_{n}\rangle \langle S_{l}\left|⊗\right|Q_{k}\rangle \langle Q_{k}|.$$
Now the meter state becomes
(10)$$\rho_{M}^{\prime }={\mathrm{Tr}}_{S}\;\rho_{\mathrm{SM}}\left(\infty \right)=\sum_{n}p_{n}^{\prime }\;|Q_{n}\rangle \langle Q_{n}|.$$
The column vector of the probabilities ${\vec{p}}^{\prime }=\left\{p_{n}^{\prime }\right\}$ is given by
(11)$${\vec{p}}^{\prime }=E^{\prime }\vec{c},$$
$E^{\prime }$ being the decoherence matrix with the elements
(12)$$E^{\prime }=\left\{E_{mn}^{\prime }\right\}=\left\{\right|\langle Q_{m}|P_{n}\rangle {|}^{2}\}.$$
The matrix $E^{\prime }$ is doubly stochastic, i.e., it satisfies
(13)$$\sum_{m}E_{mn}^{\prime }=\sum_{n}E_{mn}^{\prime }=1.$$
A comparison of the state (Equation 10) at stage 2 with the state (Equation 6) at stage 1 shows that decoherence rotates the eigenbasis of the meter state from $\left\{|P_{n}\rangle \right\}$ to $\left\{|Q_{n}\rangle \right\}$ and changes the eigenvalues from $|c_{n}{|}^{2}$ to $p_{n}^{\prime }$. Since $E^{\prime }$ is doubly stochastic, ${\vec{p}}^{\prime }$ is majorized by $\vec{c}$. As a result, the state (Equation 10) is more mixed with a higher von Neumann entropy (i.e., is more randomized) than (Equation 6), unless $\left\{|Q_{n}\rangle \right\}$ coincides with $\left\{|P_{n}\rangle \right\}$. Yet, does this randomization preclude the use of $\hat{P}$, the standard pointer, or any other pointer, for measuring the system observable $\hat{S}$?

To find out, consider that at stage 3 subsequent to stage 2, the meter undergoes a projective measurement in some basis $\left\{|R_{n}\rangle \right\}$ of an observable $\hat{R}$ arbitrarily chosen by the observer. An observation of the *m*th outcome in this basis results in the (unnormalized) post-measurement state of the system that is generally mixed. It can be written in the operator-sum representation, as
(14)$$\rho_{S,m}^{\prime }=\sum_{k}{\hat{M}}_{mk}|\psi_{S}\left(0\right)\rangle \langle \psi_{S}\left(0\right)|{\hat{M}}_{mk}^{†},$$
in terms of the Kraus operators
(15)$${\hat{M}}_{mk}=\langle R_{m}|Q_{k}\rangle \sum_{n}\langle Q_{k}|P_{n}\rangle \;|S_{n}\rangle \langle S_{n}|.$$
The measurement probabilities are then
(16)$$p_{m}=\mathrm{Tr}\;\rho_{S,m}^{\prime }=\langle \psi_{S}\left(0\right)|{\hat{E}}_{m}|\psi_{S}\left(0\right)\rangle ,$$
with
(17)$${\hat{E}}_{m}=\sum_{k}{\hat{M}}_{mk}^{†}{\hat{M}}_{mk}=\sum_{n}E_{mn}\;|S_{n}\rangle \langle S_{n}|.$$
The set of operators ${\hat{E}}_{m}$ is known as a POVM (positive operator-valued measure) [26]. Here the POVM matrix $E=\{E_{mn}\}$ is given by
(18)$$E=E^{\prime \prime }E^{\prime },$$
where
(19)$$E^{\prime \prime }=\left\{\;\right|\langle R_{m}|Q_{n}\rangle {|}^{2}\;\}.$$

The POVM operators (Equation 17) are diagonal in the basis $\left\{|S_{n}\rangle \right\}$, which means that the system observable that is measured is invariably $\hat{S}$, *irrespective of the choice of the meter basis*$\left\{|P_{n}\rangle \right\}$, $\left\{|Q_{n}\rangle \right\}$, or $\left\{|R_{n}\rangle \right\}$. This result stands contrary to the notion that the measurable system observable depends on the pointer.

Among the multitude of alternative pointer bases, the basis that conforms to Zurek’s analysis is the one that coincides with the decoherence-invariant basis,
(20)$$|R_{n}\rangle =|Q_{n}\rangle .$$
Equations (Equation 12), (Equation 18) and (Equation 19) then yield the POVM matrix
(21)$$E=\{\;|\langle R_{m}|P_{n}\rangle {|}^{2}\;\},$$
whereas Equations (Equation 14) and (Equation 15) entail a pure post-measurement state, $\rho_{S,m}^{\prime }=|\psi_{S,m}^{\prime }\rangle \langle \psi_{S,m}^{\prime }|$, where [3]
(22)$$|\psi_{S,m}^{\prime }\rangle ={\hat{M}}_{m}|\psi_{S}\left(0\right)\rangle =\sum_{n}\langle R_{m}|P_{n}\rangle c_{n}|S_{n}\rangle$$
with ${\hat{M}}_{m}=\sum_{n}\langle R_{m}|P_{n}\rangle \;|S_{n}\rangle \langle S_{n}|$.

Only under condition (Equation 20), decoherence gives rise to a nonselective measurement that does not affect the results of a subsequent selective projective measurement of the meter, since both are performed in the same basis. In all other pointer bases $\left\{|R_{n}\rangle \right\}$, decoherence affects the measurement results and/or the post-measurement states of the system, usually (partially or completely) randomizing them. However, as shown below, in most cases decoherence does not erase the information on the system, whereas in some cases it can even be beneficial for measurements.

Returning to the general case, we can rewrite (Equation 16) in the form
(23)$$\vec{p}=E\vec{c},$$
where $\vec{p}$ and $\vec{c}$ are column vectors with the components $p_{n}$ and $|c_{n}{|}^{2}$, respectively. Then $\vec{c}$ can be obtained by inverting (Equation 23),
(24)$$\vec{c}=E^{-1}\vec{p},$$
i.e., *all $|c_{n}{|}^{2}$ can be extracted from the measurement results, provided $E^{-1}$ exists*.

This inversion condition holds iff the rows (or, equivalently, columns) of E are linearly independent, or, equivalently, iff the POVM operators ${\hat{E}}_{m}$ are linearly independent. Then decoherence does not degrade the information obtainable on the system observable. The inversion condition holds iff the determinants of both $E^{\prime }$ and $E^{\prime \prime }$ are nonzero. We can then fully recover the original vector $\vec{c}$ prior to the decoherence, even though the vector $\vec{p}$ has been generally affected by decoherence.

Conversely, when all the POVM operators ${\hat{E}}_{m}$ are proportional to each other, we find that they are proportional to the identity operator with the coefficient 1/d, where *d* is the system dimensionality. This means that
(25)$$E_{mn}=1/d.$$
Inserting (Equation 25) into (Equation 23) yields that in such cases the measurement results are completely random, $p_{m}=1/d$, and reveal no information on the system.

In particular, *in the absence of decoherence* measurements provide no information on the system, when [cf. (Equation 21) and (Equation 25)]
(26)$$|\langle R_{m}|P_{n}\rangle {|}^{2}=1/d,$$
namely, the actual ($\left\{|R_{n}\rangle \right\}$) and standard ($\left\{|P_{n}\rangle \right\}$) pointer bases (Equation 5) are *mutually unbiased*.

In the presence of decoherence, Equations (Equation 12), (Equation 18), (Equation 19) and (Equation 25) imply that the decoherence can completely erase the information on the system, but only when the decoherence-invariant $\left\{|Q_{n}\rangle \right\}$ basis is mutually unbiased with either the standard pointer $\left\{|P_{n}\rangle \right\}$ or the actual-pointer $\left\{|R_{n}\rangle \right\}$ basis.

Surprisingly, decoherence is advantageous for measurements in mutually unbiased bases $\left\{|P_{n}\rangle \right\}$ and $\left\{|R_{n}\rangle \right\}$. In the absence of decoherence, a pair of such bases provides improper measurements that do not yield information on the system. However, *when decoherence occurs in the basis $\left\{|Q_{n}\rangle \right\}$, which is mutually biased with both $\left\{|P_{n}\rangle \right\}$ and $\left\{|R_{n}\rangle \right\}$, information is not erased by a measurement in the latter two bases.* This effect is counter-intuitive, since decoherence in the meter obliterates information on the system, at least partially. Nevertheless, *decoherence can turn improper measurements into proper ones*, since decoherence rotates the meter-state eigenbasis $\left\{|P_{n}\rangle \right\}$ into the basis $\left\{|Q_{n}\rangle \right\}$, which is not mutually unbiased (and thus can be dubbed mutually biased) with $\left\{|R_{n}\rangle \right\}$.

There can be intermediate cases, where the number of linearly independent POVM operators is greater than 1 but smaller than *d*. In such cases, the results of projective measurements cannot be completely reconstructed, but the POVM still provides *partial information* on the system by restricting the values of $|c_{n}{|}^{2}$.

## 4. Qubit Meter Decohered by a Bath

As an illustration of the foregoing general analysis, let us consider a two-level system (TLS) that is being measured by a qubit meter, having degenerate energy eigenstates |0〉,|1〉. Measurements in the basis of TLS energy eigenstates {|g〉,|e〉} are performed via a time-dependent TLS-meter interaction Hamiltonian of the form
(27)$$H_{\mathrm{SM}}=\left(\pi /2\right)h\left(t\right)|e\rangle \langle e|\left({\hat{I}}_{M}-{\hat{\sigma }}_{x}^{M}\right).$$
Here ${\hat{I}}_{M}$ is the identity operator of the meter, ${\hat{\sigma }}_{x}^{M}=\left|0\rangle \langle 1\right|+\left|1\rangle \langle 0\right|$, and h(t), satisfying $\int_{-\infty }^{\infty }h\left(t\right)=1$, is a smooth temporal profile of the TLS coupling to the qubit meter during the measurement that occurs in the interval centered at t=0 with duration $\tau_{M}$. A possible (but not unique) choice is the form [22]
(28)$$h\left(t\right)=\frac{1}{2\tau_{M}\cosh^{2}\left(t/\tau_{M}\right)}.$$
This form of $H_{\mathrm{SM}}$ corresponds to the controlled-not (CNOT) entangling operation [26]
(29)$${\hat{U}}_{\mathrm{CN}}=e^{-i\int_{-\infty }^{\infty }dtH_{\mathrm{SM}}\left(t\right)}.$$
If the measurement duration $\tau_{M}$ is much shorter than all other time scales, tending to the impulsive limit $\tau_{M}\to 0$, then its action is well approximated by the operator ${\hat{U}}_{\mathrm{CN}}$.

The meter–bath interaction is taken to be
(30)$$H_{\mathrm{MB}}=\left|1\rangle \langle 1\right|⊗{\hat{B}}_{1}+\left|0\rangle \langle 0\right|⊗{\hat{B}}_{0},$$
where ${\hat{B}}_{1}$ and ${\hat{B}}_{0}$ are bath operators that have orthogonal eigenstates. We may then describe stages (1) and (2) of the measurement process in Section 2 as follows:

(1) Stage 1 yields in a Schmidt-decomposed S-M correlated state:(31)$$(c_{e}|e\rangle +c_{g}|g\rangle )|0\rangle \;\to \;c_{e}|e\rangle |1\rangle +c_{g}|g\rangle |0\rangle .$$
(2) Stage 2 produces the reduced S-M density matrix. This state pertains to the standard pointer basis of M {|0〉,|1〉}, which satisfies the back-action evasion condition (Equation 7), and hence its states are invariant under the decoherence. At times much longer than the decoherence times, this state attains the diagonal form,
(32)$$\rho_{\mathrm{SM}}\to |c_{e}{|}^{2}\left|e\rangle \langle e||1\rangle \langle 1|+\right|c_{g}{|}^{2}|g\rangle \langle g||0\rangle \langle 0|.$$
(3) At stage 3, the following cases merit consideration:

(i) The actual pointer basis $\left\{|R_{n}\rangle \right\}$ coincides with $\left\{|P_{n}\rangle \right\}$ and $\left\{|Q_{n}\rangle \right\}$. The measurements of the system are then projective, and decoherence does not affect them, as shown in Section 2.

(ii) $\left\{|R_{n}\rangle \right\}$ is an arbitrary pointer basis, given by
(33)$$|R_{0}\rangle =a|0\rangle +b|1\rangle ,\;|R_{1}\rangle =b^{*}|0\rangle -a^{*}{\left|1\rangle ,\;|a\right|}^{2}+{\left|b\right|}^{2}=1.$$
In the present case, $|P_{n}\rangle =|Q_{n}\rangle$, Equations (Equation 12), (Equation 18) and (Equation 19) yield again (Equation 21), which for a qubit meter becomes
(34)$$E=\left(\begin{array}{cc} {\left|a\right|}^{2} & {\left|b\right|}^{2}\\ {\left|b\right|}^{2} & {\left|a\right|}^{2} \end{array}\right).$$

In this case, the measurement results are *the same as in the absence of decoherence*, although the meter decoherence affects the possible post-measurement states of the system, which are now mixed. In the present case, they are
(35)$$\begin{array}{ccc} & & \rho_{0}^{\prime }={\left|a\right|}^{2}|c_{g}{|}^{2}\left|g\rangle \langle g\right|+{\left|b\right|}^{2}|c_{e}{|}^{2}|e\rangle \langle e|,\\ & & \rho_{1}^{\prime }={\left|b\right|}^{2}|c_{g}{|}^{2}\left|g\rangle \langle g\right|+{\left|a\right|}^{2}|c_{e}{|}^{2}|e\rangle \langle e|. \end{array}$$

When |a|≠|b|, E has a nonzero determinant, and $\vec{c}$ in (Equation 11) can be evaluated from the POVM probabilities $p_{0},\;p_{1}$,
(36)$$|c_{g}{|}^{2}=\frac{p_{0}-{\left|b\right|}^{2}}{1-{2|b|}^{2}},\;{\left|c_{e}\right|}^{2}=\frac{p_{1}-{\left|b\right|}^{2}}{1-{2|b|}^{2}}.$$

(iii) The pointer bases with $\left|a\right|=\left|b\right|=1/\sqrt{2}$ do not provide any information on the system, since the determinant of E then vanishes. These pointer bases have the form
(37)$$\left\{(|0\rangle +e^{i\chi }\left|1\rangle \right)/\sqrt{2},\;(|0\rangle -e^{i\chi }\left|1\rangle \right)/\sqrt{2}\right\},$$
where χ is an arbitrary phase. Such pointer bases lie in the xy-plane of the Bloch sphere and are all unbiased with respect to the standard pointer basis. These pointers yield the random probabilities
(38)$$p_{0}=p_{1}=1/2.$$

Result (Equation 38) contradicts the claim that a Stern–Gerlach magnet with a field gradient in the direction *z* can measure the spin in the direction *y*. In fact, since a pointer basis of the form (Equation 37) does not provide any information on the system, it is inadequate and cannot be used for a spin measurement in any direction.

## 5. Discussion

Pointer states have been defined by Zurek [3,6,7,8] as the ones that are minimally entangled with the bath following their interaction. To find them, one quantifies the entanglement generated between the system and the bath by the von Neumann entropy obtained for the reduced density matrix of the system $\rho_{\Psi }\left(t\right)$ [initialized from $\rho_{\Psi }\left(0\right)=\left|\Psi \rangle \langle \Psi \right|$]. The pointer states are then obtained by minimizing the entropy over |Ψ〉 and demanding robustness under time variation.

When the dynamics is dominated by the system Hamiltonian, the pointer states defined as above coincide with the energy eigenstates of this Hamiltonian and conform with the view that decoherence induced by the bath *“observes”* the system and selects its pointer states. Similar conclusions apply to a meter that is coupled to a bath and measures the system.

As we have shown, *a pointer basis is not uniquely selected by decoherence*: there is a broad variety of pointer bases pertaining to a meter under the influence of a bath that still allow us to extract complete or, at least, partial information on the system. The possibility to extract the full information on the system via generalized measurement, notwithstanding the randomness of the meter observable due to decoherence, is our main result.

We note that an ideal von Neumann measurement does not reveal the phases of the superposition coefficients even without decoherence, so that the resilience of the meter basis to decoherence does not resolve the fundamental issue of quantum measurements that prompted von Neumann to introduce the projection postulate [5]. However, we may rotate the meter basis at different angles, and, for each angle, repeat the measurement on unmeasured portions of the ensemble, thereby acquiring information on the phases within an accuracy limited by the Cramer–Rao bound of estimation theory [27]. As shown by us, this bound is accessible in practice by measurements of the state decoherence combined with suitably optimized dynamical control [28]. Hence, for each rotation angle of the meter we may invoke the same considerations as the ones outlined in the present analysis. Thus, our conclusions apply in general to the acquisition of quantum information in noisy or dissipative media [29].

Zurek’s Quantum Darwinism [9,10,11] asserts that the measured information is proliferated in the environment in many copies, the observer being one of the many parts of the environment. We find, in contrast, that irrespective of the number of copies, the observer can open channels of information extraction from the system that are not constrained by the environment, by appropriately choosing the meter basis. Thus, the observer may override decoherence in almost any chosen measuring (pointer) basis. This observer’s choice can deprive resilient pointer states of their privileged status.

The present analysis gives further support to our “quantum Lamarckian” thesis [24] regarding the indispensable role of the observer. By contrast, we conclude that decoherence is neither an inevitable natural order nor a fundamental selection mechanism, but merely a reflection of the limitations of the observer’s resources. For a growing variety of systems and observables, the observer’s manipulations of the system–bath complex may render the notion of decoherence superfluous.

## Data Availability

Not applicable.

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
