# Peer review of "Does Decoherence Select the Pointer Basis of a Quantum Meter?"

_entropy, 2022, doi:10.3390/e24010106_

Round 1
Reviewer 1 Report
This paper challenges standard wisdom regarding the measurement process in quantum mechanics by showing that decoherence does not pick up a preferred pointer basis. It is highly suitable for this special issue, and disregarding the final word on the subject, it is well written and thought provoking. I recommend publication as is.
Author Response
We appreciate the reviewer's opinion, and we agree with it.
Reviewer 2 Report
The manuscript "Does Decoherence Select the Pointer Basis of a Quantum Meter" deals with the measurement model, when the environment interacting with the system-meter space (or system-apparatus, in the original Żurek's paper of 1981) induces a "measurement" on the meter, after the "measurement" of the system by the meter. The Authors pose a question of what happens if both measurements are not taken in the same basis. This leads them to the notion of alternative quantum pointers. They show what information is possible to be gained about the system, depending on the relation between different of these bases.
The result diverges from the "canonical" approach of quantum Darwinism, where the measured information is proliferated in the environment in many copies, and the observer is just one of many parts of the environment. The paper in some sense returns to the initial picture of einselection of the early Żurek's works. The Authors slightly extend the usual approach where the observer is not interacting with other parts of the environment, but in this resort, they restrict only to one of the limit cases.
The paper is interesting and in my opinion, deserves publication in the Special Issue of the Entropy journal. Still, some elements of the manuscript are not clear enough, and thus I recommend them to be clarified in the revised version.
1) The Authors use the term "back-action evasion condition" that deserves to be described in more detail.
2) It is not clear, whether H_SM is postulated to lead to (5)? Please state explicitly what constraints about H_SM are needed to be satisfied. In sec. 4 the Authors provide a specific example of CNOT interaction, but sec. 2 is general, so please specify what Hamiltonians do you consider.
3) In the paragraph before (14) the Authors write that the meter undergoes a projective measurement in some other basis R. What do Authors mean by this measurement? It is different from the measurement in basis Q, so what process chooses this basis R? Is it being einselected by some other environment, or just abstractly assumed to occur or be thought? Please elaborate on this point that seems to be crucial for providing the operational meaning of this concept.
4) The problem noted in 3), is also present further around the lines 92-94. What do the Authors mean by two consecutive projective measurements? Is it simply tracing S from S-M, and then tracing B from M-B? I guess that it is something deeper, as a measurement is more than just tracing out some of the subsystems.
5) It would improve the readability of the text if the Authors provide some summary of all introduced bases.
6) In lines 117 and 122 the term "mutually biased" bases is used. What does it mean?
7) There are many Hamiltonians leading to the CNOT interaction, adding the one chosen by the Authors in sec. 4 is rather complicated and time-dependent. What is the reason for choosing this one, not some other?
Author Response
We appreciate the penetrating comments that have prompted us to make our paper clearer.
The manuscript "Does Decoherence Select the Pointer Basis of a Quantum Meter" deals with the measurement model, when the environment interacting with the system-meter space (or system-apparatus, in the original Żurek's paper of 1981) induces a "measurement" on the meter, after the "measurement" of the system by the meter. The Authors pose a question of what happens if both measurements are not taken in the same basis. This leads them to the notion of alternative quantum pointers. They show what information is possible to be gained about the system, depending on the relation between different of these bases.
Response: We agree.
The result diverges from the "canonical" approach of quantum Darwinism, where the measured information is proliferated in the environment in many copies, and the observer is just one of many parts of the environment. The paper in some sense returns to the initial picture of einselection of the early Żurek's works. The Authors slightly extend the usual approach where the observer is not interacting with other parts of the environment, but in this report, they restrict only to one of the limit cases.
Response: We think that the significance of our results is beyond slight extension. We now clarify the principal point in the Conclusions: “ Zurek’s Quantum Darwinism asserts that the measured information is proliferated in the environment in many copies, the observer being one of the many parts of the environment. We find, in contrast, that the observer can open channels of information extraction from the system that are not constrained by the environment by appropriately choosing the meter basis.”
The paper is interesting and in my opinion, deserves publication in the Special Issue of the Entropy journal. Still, some elements of the manuscript are not clear enough, and thus I recommend them to be clarified in the revised version.
We thank the Reviewer. We have changed the paper as requested, see below:
1) The Authors use the term "back-action evasion condition" that deserves to be described in more detail.
We have now defined the term mathematically and given it a reference.
2) It is not clear, whether H_SM is postulated to lead to (5)? Please state explicitly what constraints about H_SM are needed to be satisfied. In sec. 4 the Authors provide a specific example of CNOT interaction, but sec. 2 is general, so please specify what Hamiltonians do you consider.
We have added the requested explanations in Sec. 2 in the paragraph containing Equation (4). In particular, Equation (4) provides a necessary constrain on H_SM. Sec. 4 is just an example.
3) In the paragraph before (14) the Authors write that the meter undergoes a projective measurement in some other basis R. What do Authors mean by this measurement? It is different from the measurement in basis Q, so what process chooses this basis R? Is it being einselected by some other environment, or just abstractly assumed to occur or be thought? Please elaborate on this point that seems to be crucial for providing the operational meaning of this concept.
We have now stressed that R is chosen arbitrarily by the observer. In particular, in the first paragraph of Section 3 we added the text: “Moreover, we consider a selective measurement of a meter variable R arbitrarily chosen by the observer. Generally, R commutes neither with P nor with Q.”
4) The problem noted in 3), is also present further around the lines 92-94. What do the Authors mean by two consecutive projective measurements? Is it simply tracing S from S-M, and then tracing B from M-B? I guess that it is something deeper, as a measurement is more than just tracing out some of the subsystems.
Indeed, it was unclear, and we have now provided the explanation in Sec. 2. See especially the sentence after Equation (21): “Only under condition (19), decoherence gives rise to a nonselective measurement that does not affect the results of a subsequent selective projective measurement of the meter, since both are performed in the same basis.”
5) It would improve the readability of the text if the Authors provide some summary of all introduced bases.
There are only 3 bases: P, Q and R, each described in its turn. In particular, we provided a short summary of the 3 bases in the first paragraph in Section 3.
6) In lines 117 and 122 the term "mutually biased" bases is used. What does it mean?
We have coined a new term, as now explained.
7) There are many Hamiltonians leading to the CNOT interaction, adding the one chosen by the Authors in sec. 4 is rather complicated and time-dependent. What is the reason for choosing this one, not some other?
We have now explained the condition on h(t). The example used is physical and feasible, but many others would do, as should be now clear.
Round 2
Reviewer 2 Report
The Authors made relevant changes and now the manuscript is clear enough. I recommend publishing it.